# A Subset of HOX Genes Negatively Correlates with HOX/PBX Inhibitor Target Gene Expression and Is Associated with Apoptosis, DNA Repair, and Metabolism in Prostate Cancer

**DOI:** 10.3390/genes16070824

**Published:** 2025-07-15

**Authors:** Richard Morgan, Christopher Smith, Hardev Pandha

**Affiliations:** 1School of Medicine and Biosciences, University of West London, St Mary’s Road, London W5 5RF, UK; 2Faculty of Health and Medical Sciences, University of Surrey, Guildford GU2 7XH, UK; c.a.smith@surrey.ac.uk (C.S.); h.pandha@surrey.ac.uk (H.P.)

**Keywords:** HOX, PBX, DUSP1, ATF3, HXR9, ageing

## Abstract

Background/Objectives: The *HOX* genes encode a family of homeodomain-containing transcription factors that have important roles in defining cell and tissue identity in embryonic development, but which also show deregulated expression in many cancers and have been shown to have pro-oncogenic roles. Due to their functionally redundant nature, strategies to target HOX protein function in cancer have focused on their interaction with their PBX cofactor using competitive peptides such as HXR9. HOX/PBX inhibition triggers apoptosis through a sudden increase in target gene expression, including *Fos*, *DUSP1*, and *ATF3*, which are otherwise repressed by HOX/PBX binding. Methods: We analyzed publicly available transcriptomic data in the R2 platform. Results: We show that a specific subgroup of *HOX* genes is negatively correlated with *Fos*, *DUSP1*, and *ATF3* expression in prostate cancer, and that this subgroup also shows a strong positive corelation with pathways that support tumour growth, most notably DNA repair and aminoacyl tRNA biosynthesis, and a negative correlation with genes that promote cell adhesion and prevent motility. In addition, this set of *HOX* genes strongly correlates with patient age, reflecting a previously identified progressive loss of regulation of *HOX* expression in normal peripheral blood cells. Conclusions: Our findings indicate these HOX genes may have pro-oncogenic functions in prostate cancer.

## 1. Introduction

The *HOX* genes encode a family of transcription factors that have key roles in development [1] and in adult processes that involve the renewal of cells, including the proliferation of haematopoietic stem cells and the subsequent differentiation of blood cell lineages [2]. Central to this is the ability of *HOX* genes to confer specific identities on cells and tissues, which, combined with their expression in spatial and temporal gradients in the embryo, gives them a key role in embryonic patterning [1].

In addition to their role in development, *HOX* genes are also significantly deregulated in cancer, and are frequently upregulated in most tumour types [3]. Their role in cancer is primarily pro-oncogenic, based on functional (rather than associative) studies, and includes the promotion of cell survival and proliferation, angiogenesis, metastasis, and immune invasion [3]. A major barrier to targeting HOX proteins is their functional redundancy: the four clusters of vertebrate *HOX* genes (A, B, C, and D) arose by the duplication of the ancestral *HOX* cluster found in insects, and paralog genes that share a common ancestor gene (for example, *HOXA4*, *HOXB4*, *HOXC4*, and *HOXD4*) have high levels of similarity, as do neighbouring genes within clusters (for example *HOXB4*, *HOXB5*, and *HOXB6*). Consequently, multiple HOX proteins must be knocked down or inhibited to result in a phenotypic effect [4]. One solution to this has been to target the interaction between HOX proteins encoded by *HOX* genes in paralog groups 1 through 10, and their common cofactor, PBX. PBX binding modifies the way in which HOX proteins bind to DNA and can change their intracellular distribution, causing them to relocate to the nucleus. This interaction can be inhibited using a peptide mimic of the conserved hexapeptide region in HOX proteins that mediates binding to PBX. The most frequently used inhibitor peptide, HXR9, contains a polyarginine sequence that facilitates cellular uptake, and has been shown to cause apoptosis in a wide range of solid malignancies [5].

HXR9-induced apoptosis is mediated by the loss of inhibition of three key genes—*Fos*, *Dual Specificity Phosphatase 1* (*DUSP1*), and *Activating Transcription Factor 3* (*ATF3*), all of which can trigger cell death when expressed at high levels [6,7,8,9,10]. In the case of Fos, this is primarily through the activation of the extracellular canonical apoptotic pathway via an increase in Fas Ligand (FASL) expression [11]. ATF3 also increases apoptosis by stabilising p53, leading to increased BAX expression [12]. DUSP1 inhibits Epidermal Growth Factor Receptor (EGFR) signalling through the dephosphorylation of MEK and ERK, reducing cell proliferation and survival [13]. A key role of *HOX* genes in cancer thus seems to be the repression of *Fos*, *DUSP1*, and *ATF3* expression, which in turn inhibits apoptosis, and correspondingly *DUSP1* [14] and *ATF3* [15] have both previously been identified as potential tumour suppressor genes in prostate cancer, although the role of *Fos* in this context is less clear.

In this bioinformatic analysis, we have identified *HOX* genes that show a negative correlation with *Fos*, *DUSP1*, and *ATF3* expression in prostate tumours. The expression of this subset of *HOX* genes is also strongly correlated with that of genes involved in DNA repair, metabolism, and motility, indicating that they have additional pro-oncogenic roles. Furthermore, they show a significant increase in tumour expression in older patients, reflecting previous findings that *HOX* genes become progressively more highly expressed with age in some individuals.

## 2. Methods

### 2.1. Microarray Data

The dataset used in this analysis is from the study by Ross-Adams et al., in which 103 samples from men with prostate cancer and 99 matched benign samples were profiled using the Illumina HT12v4 expression array to provide transcriptomic data [16]. The clinical characteristics of this cohort are described in detail elsewhere, but the key clinical parameters included are age at diagnosis (range, 41–93 years), biochemical relapse, clinical stage, Gleason grade, extra capsular extension, positive surgical margins, PSA at diagnosis, and total follow up time (range, 2–67 months). Bead-level data were processed to remove spatial artefacts and then log2-transformed and quantile-normalised. Probes that did not match their intended genomic location or which mapped to genomic areas that were not useful for transcription studies were excluded from the analysis. For validation, the analyses were repeated on a second, independent dataset [17]. This contains transcriptomes from 131 primary prostate tumours, 29 matched normal tissues, and 19 metastases from patients with a mean age of 58 years at diagnosis (range, 37–83 years).

### 2.2. Data Analysis

Data analysis was performed using the R2 platform (R2: Genomics Analysis and Visualization Platform (http://r2.amc.nl, accessed on 1 July 2025)). In order to generate the HOX_DFA3 gene set, individual gene vs. gene correlations were performed for each of the putative target genes (*Fos*, *DUSP1*, and *ATF3*) with each of the *HOX* genes in paralog groups 1–10 (30 *HOX* genes in total) using the “Correlate 2 genes” function, and *HOX* genes that negatively correlated with one or more of the target genes with *p* < 0.05 and r < −0.1 were included in the HOX_DFA3 group. For gene set vs. gene set correlations, R2 was used to perform a regression analysis for the mean value of the genes in the HOX_DFA3 gene set against other defined gene sets (KEGG) for each sample, with transformation based on the Z-score. For correlations between the HOX_DFA3 gene set and patient age at diagnosis, the Track (#) vs. Genesets Correlations function was used, again with the Z-score transformation.

For individual *HOX* genes, we compared expression with specific clinical and pathological criteria using the gene vs. track function and a log2 transformation, using a 1-way ANOVA to test for significance (*p* < 0.0033, applying the Bonferroni correction to account for the 14 different *HOX* genes in the HOX_DFA3 gene set).

## 3. Results

### 3.1. DUSP1 and FOS Are Both Significantly More Highly Expressed in Benign Prostate Tissue Compared to Tumour Tissue

Given the potential tumour suppressor roles of *DUSP1*, *FOS*, and *ATF3*, we compared the expression of these genes in prostate tumours and benign prostate tissue. This reveals that all three of these genes are expressed more highly in benign tumours (Figure 1), although only significantly so in the case of *DUSP1* (*p* = 2.02 × 10^−12^) and *Fos* (*p* = 1.18 × 10^−7^).

### 3.2. A Subset of HOX Genes Negatively Correlate with DUSP1, Fos, and ATF3 Expression

As described above, *DUSP1*, *Fos*, and *ATF3* are chosen as a starting point because previous studies showed that their upregulation mediated the apoptosis induced by the HOX/PBX antagonist HXR9. Hence, the first part of this study seeks to identify *HOX* genes that show a negative association with the expression of *ATF3* and/or *DUSP1* and/or *Fos* in order to determine a putative set of *HOX* genes that may be responsible for the cytotoxicity of HXR9. To establish which *HOX* genes showed an mRNA expression that was negatively correlated with *Fos*, *DUSP1*, or *ATF3*, we compared their individual expression to that of each *HOX* gene in paralog groups 1–10 through a regression analysis (Table 1), with a cutoff value of *p* < 0.05 and r < −0.1. This indicates that 14 of the 30 *HOX* genes in these paralog groups are significantly correlated with the expression of at least one of *Fos*, *DUSP1*, or *ATF3*, and that 5 *HOX* genes (*HOXA10*, *HOXC4*, *HOXC6*, *HOXC9*, and *HOXD8*) are significantly correlated with the expression of all three (Figure 2).

On this basis, we can define a subset of *HOX* genes which are negatively correlated with *DUSP1* and/or *Fos* and/or *ATF3*—*HOXA6*, *HOXA9*, *HOXA10*, *HOXB3*, *HOXB5*, *HOXB6*, *HOXB7*, *HOXC4*, *HOXC5*, *HOXC6*, *HOXC9*, *HOXD1*, *HOXD3*, and *HOXD8*. For subsequent analyses, these are treated as a single gene set, referred to as DFA3_HOX. We then performed a correlation analysis between DFA3_HOX and *DUSP1*, *Fos*, and *ATF3* as a single gene set, referred to as DUSP1_FOS_ATF3 subsequently. This reveals a very strong negative association between DFA3_HOX and their putative target genes, DUSP1_FOS_ATF3, with an R-value of −0.469 and a *p*-value of 2.70 × 10^−12^ (Figure 3).

### 3.3. DFA3_HOX Expression Correlates with the Genes Involved in Cell Adhesion, DNA Repair, Cell Metabolism, and Translation

Next, we looked to see if the *HOX* genes in DFA3_HOX had significant associations with other gene groups in the KEGG pathways of defined cellular processes using an R-value cutoff of <−0.2 and >0.2. This reveals a significant negative correlation with the expression of genes involved in cell adhesion (Figure 4, Appendix A), and a significant positive correlation with the expression of genes involved in DNA repair (Figure 5, Appendix A), translation (Figure 6, Appendix A), and cellular metabolism (Figure 7, Appendix A). Amongst these, the strongest correlations were between DFA3_HOX and genes involved in the Fanconi Anemia Pathway (r = 0.407, *p* = 2.49 × 10^−9^), base excision repair (r = 0.382, *p* = 2.48 × 10^−8^), and aminoacyl tRNA synthesis (r = 0.425, *p* = 4.02 × 10^−10^).

In addition to the relationships between gene sets, we looked to see if there was a relationship between the DFA3_HOX gene set and clinical characteristics of the patient cohort. This revealed only one potentially significant positive correlation, which was between DFA_HOX3 expression and patient age at diagnosis, with an R-value of 0.22 and *p*-value of 0.014 (Figure 8).

### 3.4. Individual HOX_DFA3 Gene Expression in Tumour vs. Normal Tissue

We also assessed whether any of the individual genes in the HOX_DFA3 gene set showed a significantly different level of expression between normal prostate tissue and tumours, with a cutoff *p*-value of 0.0033 (applying the Bonferroni correction with respect to the 14 HOX genes in this group). Comparing benign to tumour samples, *HOXA6* (*p* = 4.26 × 10^−5^), *HOXB6* (*p* = 0.001), *HOXC5* (*p* = 4.69 × 10^−5^), *HOXC6* (*p* = 3.02 × 10^−28^), and *HOXC9* (*p* = 0.0001) all showed significantly greater expression in tumour samples, with *HOXC6* having by far the greater fold difference (Figure 9). In addition, we performed a heat map analysis for all the *HOX* genes in HOX_DFA3, which reveals that most normal prostate tissue samples have significantly lower expression of these genes compared to tumour tissue (Appendix A).

### 3.5. Validation with Independent Datasets

To determine whether the findings from our analysis are applicable beyond the Ross-Adams dataset, we repeated the analyses on data from a second, independent study. This second dataset contains transcriptomes from a total of 131 primary prostate tumours, 19 metastases, and 29 normal prostate tissue samples [17]. This revealed similarly significant relationships between HOX_DFA3 and the other gene sets described above (Table 2).

## 4. Discussion

In this analysis of *HOX* gene expression in prostate tumours and matched benign tissue, we have identified a subset of 14 *HOX* genes, DFA3_HOX, the expression of which is negatively correlated with *ATF3*, *DUSP1*, and *Fos*. These three genes have previously been identified as targets of HOX/PBX-mediated transcriptional repression, which in turn is a key survival mechanism in at least some types of cancer cells, as they promote apoptosis through both intrinsic and extrinsic pathways. This strong negative correlation between DFA3_HOX and *ATF3*, *DUSP1*, and *Fos*, therefore, supports the hypothesis based on in vitro data and mouse models that *HOX* genes have an important role in preventing apoptosis.

Our analysis also revealed that this subset of *HOX* genes has a strong negative correlation with genes involved in cell adhesion, specifically those that encode proteins in gap, focal adhesion, and adherens junctions. This concurs with previous studies that have shown a relationship between *HOX* genes and cell migration, and, in particular, with the epithelial to mesenchymal transition (EMT) [3]. Multiple *HOX* genes are involved in EMT, and there is evidence of redundancy in their function—for example, *HOXB7*, *HOXB8*, *HOXC8*, *HOXC10*, and *HOXD3* all repress *E-cadherin* expression [18,19,20,21,22,23,24,25], and similarly, *HOXB8* and *HOXD3* promote *N-cadherin* expression [20,21,26,27].

The DFA3_HOX genes also show a strong positive correlation with genes involved in pro-oncogenic processes—DNA repair, translation, and cell metabolism. Multiple *HOX* genes have previously been shown to have roles in DNA repair, for example, HOXB7 directly interacts with components involved in double-stranded break repair (DSB) [28], and *HOXB9* [29] and *HOXA10* [30] also promote DSB. Conversely, there do not seem to be any previous studies identifying HOX transcription factors as regulators of genes involved in translation and cell metabolism, although the generally reported pro-oncogenic role for *HOX* genes is consistent with the association reported here [3].

In addition to these associations, we also identified a significant positive correlation between patient age at diagnosis and expression of DFA3_HOX genes with a regression value of 0.22 (*p* = 0.014). A potential relationship between *HOXA9* expression in normal peripheral blood cells and age has previously been reported [31], in which *HOXA9* expression was relatively low until the age of 60 years, at which point there was a significant increase in mean expression (*p* < 0.0001). Subsequent studies have indicated that promoter methylation may play a role in age-related changes in *HOX* gene expression. For example, *HOXA5* (which is considered to have a tumour-suppressor rather than pro-oncogenic role) undergoes increasing promoter methylation with age and a corresponding loss of expression at both the RNA and protein level [32], and in a study of muscle cells from young (27 ± 4.4 years) and old (83 ± 4 years) healthy volunteers, significantly increased methylation of regulatory regions was shown for multiple *HOX* genes [33].

Taken together, these findings support the previously reported pro-oncogenic role for *HOX* genes [3], which includes, for the DFA3_HOX subgroup, the suppression of pro-apoptotic genes and genes that inhibit cell movement, and the transcriptional activation of the genes involved in DNA repair, translation, and metabolism. Several of these potential regulatory interactions, especially with respect to translation and metabolism, have not previously been reported, and their mechanistic basis requires further investigation.

## Figures and Tables

**Figure 1 genes-16-00824-f001:**
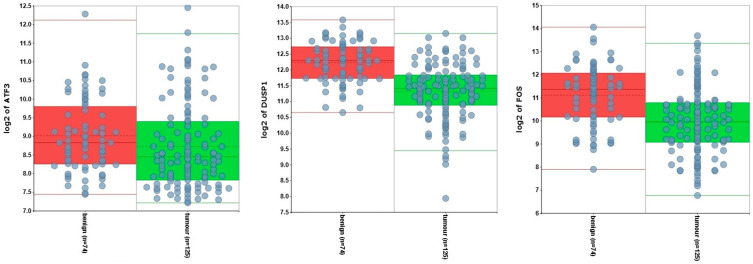
*ATF3*, *DUSP1*, and *FOS* expression in normal prostate tissue (red) and prostate tumour (green). Data are presented in a box and whiskers plot showing the median (centre of the box), IQR, and range, with each dot representing an individual sample.

**Figure 2 genes-16-00824-f002:**
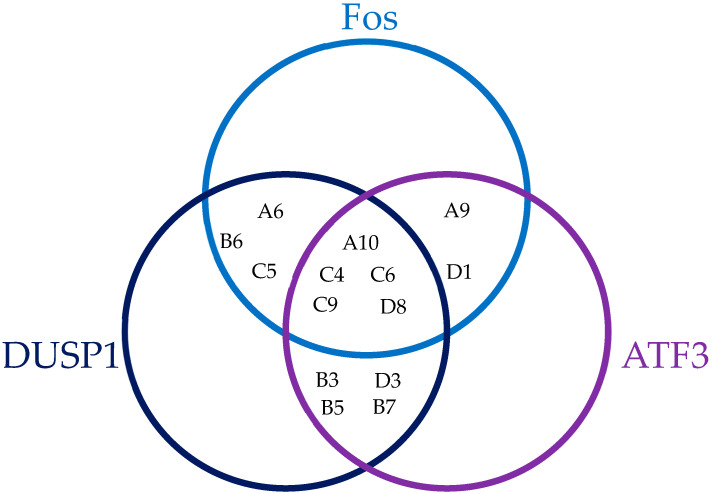
Venn diagram showing *HOX* gene expression that significantly negatively correlates with the expression of *Fos*, *DUSP1*, and/or *ATF3* in the Ross-Adams dataset.

**Figure 3 genes-16-00824-f003:**
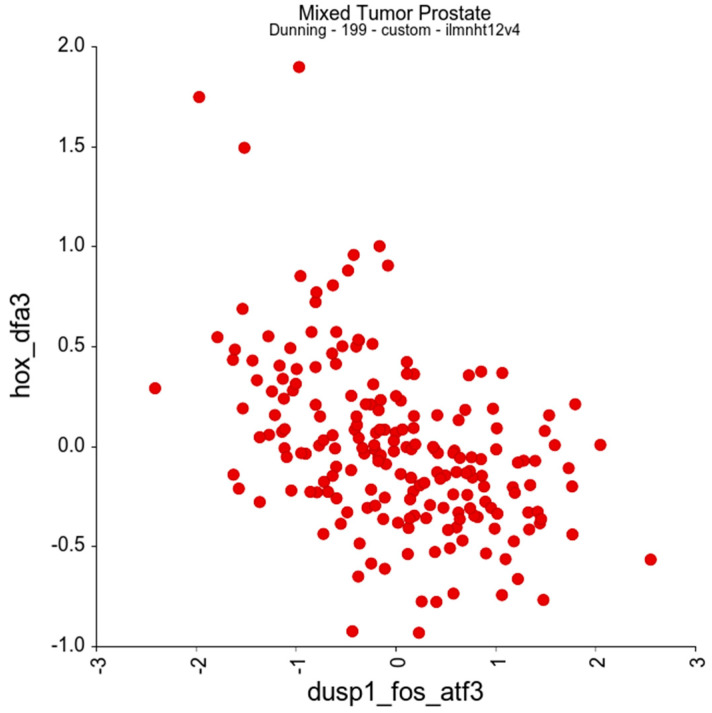
The expression of the *DUSP1*/*Fos*/*ATF3* gene set shows a significant negative correlation with a subset of *HOX* genes, DFA3_HOX (r = −0.469, *p* = 2.70 × 10^−12^).

**Figure 4 genes-16-00824-f004:**
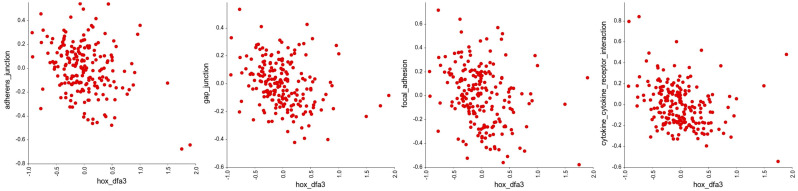
Correlation analysis of the *HOX* genes in DFA3_HOX with gene groups in the KEGG pathways related to cell adhesion. Of these, significant negative correlations were found for the genes involved in adherens junctions (r = −0.305, *p* = 1.17 × 10^−5^), gap junctions (r = −0.285, *p* = 4.41 × 10^−5^), focal adhesion (r = −0.267, *p* = 1.40 × 10^−4^), and cytokine–cytokine receptor interactions (r = −0.276, *p* = 7.80 × 10^−5^).

**Figure 5 genes-16-00824-f005:**
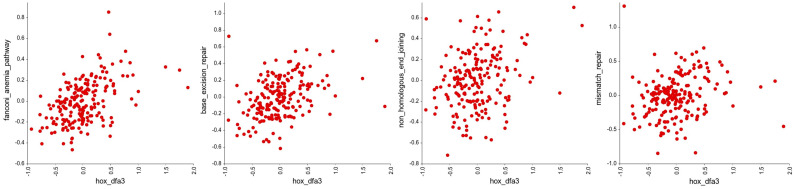
Correlation analysis of the *HOX* genes in DFA3_HOX with gene groups in the KEGG pathways related to DNA repair. Of these, significant positive correlations were found for the genes involved in the Fanconi anaemia pathway (r = 0.407, *p* = 2.49 × 10^−9^), base excision repair (r = 0.382, *p* = 2.48 × 10^−8^), non-homologous end-joining repair (r = 0.270, *p* = 1.15 × 10^−4^), and mismatch repair (r = 0.213, *p* = 0.00251).

**Figure 6 genes-16-00824-f006:**
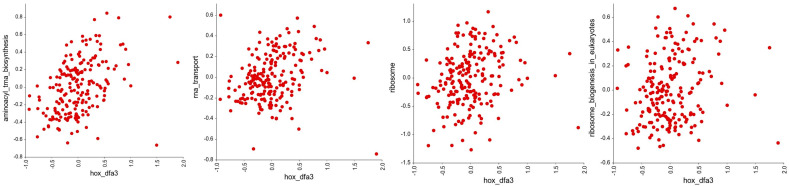
Correlation analysis of the *HOX* genes in DFA3_HOX with gene groups in the KEGG pathways related to translation. Of these, significant positive correlations were found for the genes involved in aminoacyl tRNA biosynthesis (r = 0.425, *p* = 4.02 × 10^−10^), RNA transport (r = 0.228, *p* = 0.00223), ribosomes (r = 0.204, *p* = 0.0038), and ribosome biogenesis (r = 0.184, *p* = 0.00932).

**Figure 7 genes-16-00824-f007:**
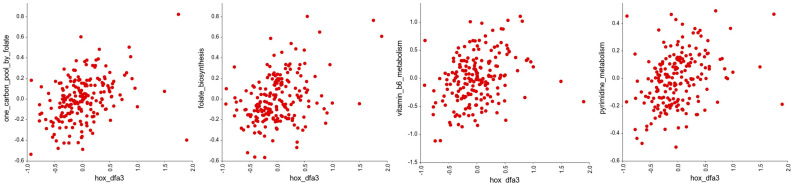
Correlation analysis of the *HOX* genes in DFA3_HOX with gene groups in the KEGG pathways related to metabolism. Of these, significant positive correlations were found for the genes involved in the folate one-carbon pool (r = 0.369, *p* = 8.36 × 10^−8^), folate biosynthesis (r = 0.352, *p* = 3.31 × 10^−7^), vitamin B6 metabolism (r = 0.350, *p* = 3.97 × 10^−7^), and pyrimidine metabolism (r = 0.326, *p* = 2.60 × 10^−6^).

**Figure 8 genes-16-00824-f008:**
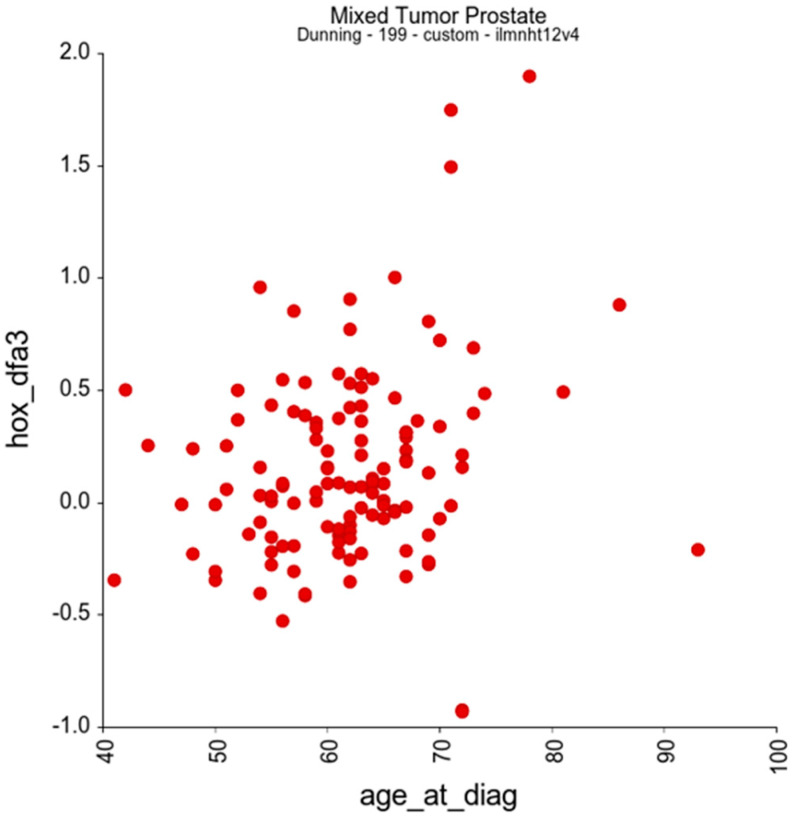
Correlation analysis of the expression of DFA3_HOX genes in prostate tumours with patient age at diagnosis for the prostate cancer expression set, revealing a significant positive association (r = 0.220, *p* = 0.014).

**Figure 9 genes-16-00824-f009:**
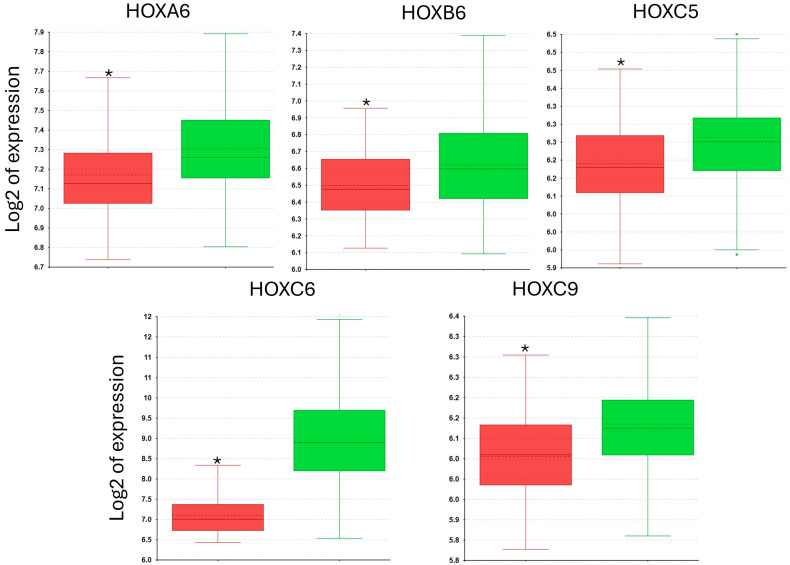
*HOXA6*, *HOXB6*, *HOXC5*, *HOXC6*, and *HOXC9* expression in normal prostate tissue (red) and prostate tumour (green). Data are presented in a box and whiskers plot showing the median (centre of the box), IQR, and range (*, 1-way ANOVA, *p* < 0.0033).

**Table 1 genes-16-00824-t001:** *HOX* genes with a significant negative correlation with *Fos*, *ATF3*, and *DUSP1* expression in the prostate array data described in the Ross-Adams study.

	*ATF3*		*DUSP1*		*Fos*	
	*p*-Value	R-Value	*p*-Value	R-Value	*p*-Value	R-Value
*HOXA6*	-	-	0.00207	−0.217	0.05	−0.139
*HOXA9*	0.00135	−0.226	-	-	0.05	−0.133
*HOXA10*	0.000383	−0.249	0.023	−0.161	0.000436	−0.247
*HOXB3*	0.03	−0.154	0.05	−0.123	-	-
*HOXB5*	0.012	−0.179	0.000777	−0.236	-	-
*HOXB6*	-	-	0.00304	−0.209	0.037	−0.148
*HOXB7*	0.041	−0.145	0.013	−0.176	-	-
*HOXC4*	0.046	−0.141	0.00000000273	−0.406	0.00000785	−0.311
*HOXC5*	-	-	0.045	−0.142	0.05	−0.123
*HOXC6*	0.024	−0.160	0.00000000889	−0.394	0.000000452	−0.349
*HOXC9*	0.05	−0.138	0.000134	−0.267	0.00841	−0.186
*HOXD1*	0.000151	−0.265	-	-	0.025	−0.159
*HOXD3*	0.037	−0.148	0.000406	−0.248	-	-
*HOXD8*	0.016	−0.170	0.00696	−0.191	0.00885	−0.185

**Table 2 genes-16-00824-t002:** Testing of associations in an additional, independent dataset. For the dataset used for validation (right-hand column), green indicates a significant association with the same direction of correlation compared to the dataset from the Ross-Adams study. Values in bold indicate a greater R-value and lower *p*-value than for the Ross-Adams dataset.

Tested Association: HOX_DFA3 vs	Ross-Adams (Analysed in This Paper; [16])	Taylor [17]
DUSP1_FOS_ATF3	r = −0.469, *p* = 2.70 × 10^−12^	r = −0.232, *p* = 6.22 × 10^−6^
Fanconi anaemia pathway	r = 0.407, *p* = 2.49 × 10^−9^	r = 0.279, *p* = 4.82 × 10^−8^
Base excision repair	r = 0.382, *p* = 2.48 × 10^−8^	** r = 0.431, *p* = 3.32 × 10^−18^ **
Aminoacyl tRNA synthesis	r = 0.425, *p* = 4.02 × 10^−10^	r = 0.158, *p* = 0.0023
Pyrimidine metabolism	r = 0.326, *p* = 2.60 × 10^−6^	r = 0.277, *p* = 6.15 × 10^−8^
Folate one-carbon pool	r = 0.369, *p* = 8.36 × 10^−8^	r = 0.231, *p* = 7.02 × 10^−6^

## Data Availability

All of the data is publicly available through the R2 platform (http://r2.amc.nl).

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
