# Peer review of "A Subset of HOX Genes Negatively Correlates with HOX/PBX Inhibitor Target Gene Expression and Is Associated with Apoptosis, DNA Repair, and Metabolism in Prostate Cancer"

_genes, 2025, doi:10.3390/genes16070824_

Round 1
Reviewer 1 Report
Comments and Suggestions for Authors
- In the introduction section, the authors need to take a step back and explain the rationale for studying the “target genes” - Fos, DUSP1, and ATF3. As there are many pathways and key genes associated with the development of prostate cancer, please provide the context for why apoptosis is selected, and why these three genes (vs. other key genes) are used as the “target genes” in this study. Also, the literatures cited here are not prostate-cancer related (e,g., ref 11 is in melanoma, ref 12 is in TCL, and ref 13 is in NSCLC and CRC). Please provide strong rationale and context of why these genes are studied and why this is an important topic in prostate cancer
- At the beginning of the result section, the authors should also assess the baseline expression of the “target genes” in this study - i.e., the baseline expression of Fos, DUSP1, and ATF3. Are their expression levels dysregulated in prostate cancer tissue vs. benign tissue?
- The primary assay that was being analyzed was the microarray-based transcriptome profile. Also, most figures based on correlation analyses. Although these findings are interesting, they are not strong enough to support future studies in identifying a new critical target, unveiling which factors among the HOX family are most associated with prostate cancer development. The molecular bio expertiments to validate and demonstrate the expression profile of HOX genes and the expression of Fos, ATF3 and DUSP1 are needed.
Here are some minor points:
- For figures, please make sure all the legends and tags are visible - e.g., the labels on figure 9 are overlapping and cannot be recognized
Author Response
Many thanks for your kind consideration of our manuscript.
- These genes are chosen as a starting point because previous studies showed that their upregulation mediated the apoptosis induced by the HOX/PBX antagonist HXR9. The first part of our study is to identify HOX genes that show a negative association with the expression of ATF3 and/or DUSP1 and/or Fos in order to determine a putative set of HOX genes that could be responsible for the cytotoxicity of HXR9. This explanation is now included in the Results section (highlighted in red). Specific references for the role of DUSP1 and ATF3 in prostate cancer are now included (Introduction, highlighted in red).
- Thank you for this very helpful suggestion, the analysis is included in the revised paper (highlighted in red, figure 1).
- Thank you, this is an important point. The study is an associative one, as you point out, however we are not suggesting that the group of HOX genes identified is a potential therapeutic target (and of course direct experimental evidence would be needed to support this). Instead, we are seeking to establish whether the observations previously made in cell culture / mouse studies potentially correspond to patterns of gene expression seen in clinical samples, and thus we would argue that the use of this transcriptomic data is appropriate.
Minor point
- Thank you for pointing this out. All of the images can be provided as separate JPEG files.
Reviewer 2 Report
Comments and Suggestions for Authors
This manuscript describes a simple computer (bioinformatics) based interrogation of 2 publicly available prostate cancer databases. The study has correlated expression of several HOX genes with expression of genes involved in various other pathways and the results are consistent with findings from other, more comprehensive functional studies. The analyses performed are basic, and because most of the results have already been investigated previously, overinterpretation of the data is limited. The authors could consider the following points.
- The line numbering has been corrupted and only appears in the reference section.
- The formatting in some figures could be improved (e.g. Figure 9). Unless raw data will be added as supplementary data, all labels should be readable.
- Due to the increasing use of AI summary tools, wording in the manuscript needs to be precise so that study results are not misinterpreted. For example, in the final paragraph of the Introduction where the authors have summarised their work, use of more specific words/phrases would be beneficial. (e.g. (a) In this bioinformatic analysis, we have identified… [add ‘bioinformatic’], (b) Expression of this subset of HOX genes is also strongly correlated with expression of genes involved in… [specify ‘expression of’]).
- On what basis are the authors concluding that HOX gene expression is “deregulated” with age? Previous studies have identified age-related global changes in HOX gene expression, in some cases in association with age-associated changes in DNA methylation. “Deregulated” implies that this is abnormal. Is this what the authors intended, or is the increase in HOX gene expression a normal process of ageing?
- When the authors have investigated HOX gene expression with age, were they using HOX gene expression in the prostate cancer or the benign prostatic tissue? This should be defined.
- To investigate HOX gene expression in tumour vs benign prostatic tissue, could the authors (additionally) look at fold changes between the matched benign and malignant specimens in order to specifically profile the tumour-associated changes in HOX gene expression? (At present, the non-malignant prostate and prostate tumour specimens are only analysed as independent groups).
- Have the authors accounted for the different cellular composition of the benign and malignant prostate tissues when interpreting their analyses (e.g proportion of epithelial vs non-epithelial (stromal cells, lymphocytes, etc)?
Author Response
Many thanks for your kind consideration of our manuscript.
- The line numbering was not in the submitted manuscript and I think it must have been introduced by the editorial team.
- Thank you for pointing this out. All the images can be provided as separate JPEG files.
- AI was not used in writing this manuscript, however we thank you for your suggestions and have made the required changes (highlighted in blue).
- Thank you, this is an important point and an interesting one. The study we refer to on HOXA9 expression in normal PBMCs showed that expression was low in cells from individuals aged <40 years. Over 60 years of age some (apparently healthy) individuals showed highly elevated HOXA9 expression in these cells, whilst others did not. However, you are correct in that this does not prove that it isn’t simply part of a normal ageing process. We have therefore changed the text in the introduction to “…reflecting previous findings that HOX genes become progressively more highly expressed with age in some individuals” (highlighted in blue).
- This was in tumor tissue – the legend for Figure 8 has been amended to make this clear (change shown in blue).
- Thank you, we have now included a heat map showing expression of all the HOX genes in the HOX_DFA3 set in individual tumour and benign samples. This is included as a supplementary figure as it requires magnification to see useful detail. Corresponding text has been added to the Results section (highlighted in blue).
- This is a valid point, certainly. However, I think it is generally accepted that adjacent normal tissue will contain these different cell types in variable proportions, just as tumours typically contain some non-tumour cells, which is an acknowledged limitation when working with primary tissue rather than cell lines.
Round 2
Reviewer 1 Report
Comments and Suggestions for Authors
Thanks for the response and revision. As the objective of this study is bioinformatics analysis, the authors should perform more analysis with the microarray data instead of adding validation experiment; For pathway analysis, more visualization is needed to support the conclusion.
For KEGG pathway analysis, please add the enrichment analysis or gene network analysis that shows the expression profile (upregulation / downregulation) of individual genes in a pathway.
Author Response
Thank you again for your comments and advice. We have now performed heatmap analysis for all the KEGG pathways identified in the study (Supplementary Figures 1-4), showing the upregulation and downregulation of each gene. Additional changes to the text including new figure legends are highlighted in purple.
Round 3
Reviewer 1 Report
Comments and Suggestions for Authors
Thanks for adding the supplementary figures. I do not have additional comments.